# The malaria parasite PP1 phosphatase controls the initiation of the egress pathway of asexual blood-stages by regulating the rounding-up of the vacuole

Marie Seveno[1], Manon N. Loubens[1], Laurence Berry[1], Arnault Graindorge[1], Maryse Lebrun[1], Catherine Lavazec[2], Mauld H. Lamarque[1]*

1 LPHI, UMR 5294 CNRS/UM–UA15 Inserm, Université de Montpellier, Montpellier, France, 2 INSERM U1016, CNRS UMR8104, Université Paris Cité, Institut Cochin, Paris, France

* mauld.lamarque@umontpellier.fr

## Abstract

A sustained blood-stage infection of the human malaria parasite *P. falciparum* relies on the active exit of merozoites from their host erythrocytes. During this process, named egress, the infected red blood cell undergoes sequential morphological events: the rounding-up of the surrounding parasitophorous vacuole, the disruption of the vacuole membrane and finally the rupture of the red blood cell membrane. These events are coordinated by two intracellular second messengers, cGMP and calcium ions ($Ca^{2+}$), that control the activation of their dedicated kinases, PKG and CDPKs respectively, and thus the secretion of parasitic factors that assist membranes rupture. We had previously identified the serine-threonine phosphatase PP1 as an essential enzyme required for the rupture of the surrounding vacuole. Here, we address its precise positioning and function within the egress signaling pathway by combining chemical genetics and live-microscopy. Fluorescent reporters of the parasitophorous vacuole morphology were expressed in the conditional *Pf*PP1-iKO line which allowed to monitor the kinetics of natural and induced egress, as well as the rescue capacity of known egress inducers. Our results underscore a dual function for PP1 in the egress cascade. First, we provide further evidence that PP1 controls the homeostasis of the second messenger cGMP by modulating the basal activity of guanylyl cyclase alpha and consequently the PKG-dependent downstream $Ca^{2+}$ signaling. Second, we demonstrate that PP1 also regulates the rounding-up of the parasitophorous vacuole, as this step is almost completely abolished in *Pf*PP1-null schizonts. Strikingly, our data show that rounding-up is the step triggered by egress inducers, and support its reliance on $Ca^{2+}$, as the calcium ionophore A23187 bypasses the egress defect of *Pf*PP1-null schizonts, restores proper egress kinetics and promotes the initiation of the rounding-up step. Therefore, this study places the phosphatase PP1 upstream of the cGMP-PKG signaling pathway, and sheds new light on the regulation of rounding-up, the first step in *P. falciparum* blood stage egress cascade.

**Data Availability Statement:** All relevant data are within the manuscript and its Supporting Information files.

**Funding:** M.H.L is funded by the Agence Nationale de la Recherche (ANR-21-CE15-0043) (https://anr.fr/) and M.L. is funded by the Laboratoire d'Excellence (LabEx) (ParaFrap ANR-11-LABX-0024) (https://www.labex-parafrap.fr/fr/). The funders had no role in study design, data collection and analysis, decision to publish, or preparation of the manuscript.

**Competing interests:** The authors have declared that no competing interests exist.

## Author summary

Malaria caused by *Plasmodium falciparum* infections remains a major human threat in endemic countries. Its proliferation within the host relies on the iteration of red blood cell invasion, multiplication and release of newly formed parasites in the blood circulation. This last step, named egress, is tightly regulated by a signaling pathway controlled by phospho-regulation. The phosphatase PP1 is a conserved pleiotropic enzyme that regulates various biological processes in mammals and controls the replication and egress mechanisms in *P. falciparum*. Indeed, PP1-depleted parasites are unable to egress from the erythrocytes and remain trapped within a vacuole in the host cell. Here, using fluorescent reporters of the surrounding vacuole, and pharmacological inducers of the egress pathway, we analyzed natural and induced egress by time-lapse video-microscopy. Our results underscore a dual function of PP1 during egress and identify the phosphatase as an early regulator of this essential process.

## Introduction

Parasites belonging to the Apicomplexa phylum are the causative agents of many diseases of human and veterinary importance, such as *Plasmodium falciparum* responsible for human malaria, *Toxoplasma gondii* the etiologic agent of toxoplasmosis, or *Cryptosporidium* that accounts for gastrointestinal diseases mainly in developing countries. Malaria still remains a public health concern with over 0.6 million deaths reported in 2022 by WHO, of which 90% can be accounted for by *P. falciparum* [1]. This parasite alternates its life cycle between the Anopheles mosquito vector and the human host [2]. The infection starts upon inoculation of sporozoites during the blood meal of an infected female mosquito. Following an asymptomatic replication in the liver, thousands of new invasive stages named merozoites are released in the blood circulation, where they initiate the symptomatic phase of the disease, namely the asexual blood stage [3]. Merozoites invade red blood cells (RBCs) and during the process become enclosed inside a vacuole derived from the host cell membrane and known as the parasitophorous vacuole (PV) [4,5]. The replication that ensues leads to the formation of up to 32 merozoites [6] that exit from the host cell by an active mechanism termed egress, to be once again released in the blood circulation. Due to its pathological relevance and ability to be recapitulated *in vitro* [7], the asexual blood-stage of *P. falciparum* has been the focus of most of the studies on malaria.

As an obligate intracellular parasite, *P. falciparum* blood-stage merozoites must exit the host cell to allow for parasite dissemination and expansion. Egress is a multi-step mechanism that is finely regulated, ensuring the timely release of invasive parasites in the blood circulation. It starts with the poration of the PV membrane (PVM) [8] several minutes before a characteristic change in the morphology of the PV, known as the rounding-up, where the PV takes a more spherical shape without swelling and in a calcium-dependent manner [9–11]. Then, the PV abruptly ruptures, followed by RBC membrane (RBCM) poration and subsequently RBCM opening, curling and buckling to favor parasite dissemination [12]. The signaling cascade leading to egress has been largely focused on protein kinase G (PKG), a cGMP-dependent kinase essential for the disruption of the surrounding membranes [13,14]. cGMP homeostasis is maintained by the opposite activities of guanylyl cyclases (GCs) for synthesis and phosphodiesterases (PDEs) for hydrolysis. The *Plasmodium* genome encodes 2 GCs, GCα being the relevant one as it comes to asexual blood-stage egress [15,16]. In *T. gondii* and *Plasmodium*, GCs

form multimeric protein platforms by their association with several protein cofactors including cell division control 50 (CDC50), Unique GC Organizer (UGO) and Signaling Linking Factor (SLF) [17–20]. Elevation of cGMP concentrations stimulates PKG, which in turn activates phosphatidyl inositol phospholipase C (PI-PLC), thereby generating diacylglycerol (DAG) and inositol triphosphate ($IP_3$) [21,22]. In mammalian cells, $IP_3$ binds to $IP_3$ receptors localized at the membrane of the endoplasmic reticulum (ER), triggering the release of calcium ions ($Ca^{2+}$) into the cytoplasm [23]. In *Plasmodium*, despite the lack of clear $IP_3$ receptor orthologs, PKG-mediated $Ca^{2+}$ mobilization takes place [22,24,25], a phenomenon dependent on ICM1, a multipass membrane protein that physically interacts with PKG [26]. Calcium effectors such as calcium-dependent protein kinase 5 (CDPK5) then triggers the discharge of at least two parasitic secretory organelles, the exonemes and micronemes [27,28]. The release of the exonemal protein SUB1 into the PV allows the processing of other parasitic proteins, including the cysteine proteases of the SERA family and merozoite surface proteins [14,29–32]. SUB1-processed targets may assist RBC cytoskeleton destabilization [30,31], while a PV-resident phospholipase LCAT has been recently implicated in the disruption of both the PVM and RBCM [33].

The egress signaling pathway is regulated by phosphorylation events and we previously identified the first phosphatase of the parasite involved in its regulation [34]. PP1 is an evolutionary ancient phosphatase highly conserved in eukaryotes. In mammals, it regulates many cellular processes *via* its interactions with hundreds of regulatory subunits, thus forming so called holoenzymes [35], a versatile trait conserved in *Plasmodium* [36]. In *P. falciparum*, we showed that PP1 is essential during the asexual blood-stage, due to a dual function during schizogony and during egress [34]. Secretion of exonemes and micronemes was abrogated in *Pf*PP1-null parasites, thereby preventing PVM rupture. This phenotype was correlated with hyperphosphorylation of GCα and lower cGMP levels, suggesting that PP1-dependent dephosphorylation of GCα was of crucial importance to properly activate the PKG signaling pathway.

In this study, we extended our understanding of PP1 function during egress. Based on chemical genetics using pharmacological inducers of the egress pathway, combined with live-microscopy of *Pf*PP1-iKO parasites egress events, we document at an unprecedented scale the kinetics of natural and induced egress. By comparing the effects of egress inducers on control and *Pf*PP1-null parasites, we validate the cGMP-PKG cascade as an important target of the phosphatase. More importantly, we identify PV rounding-up as the primary step regulated by PP1, placing the phosphatase at the initiation of the egress cascade.

## Results

### $Ca^{2+}$ mobilization by an ionophore fully rescues the egress defect of *Pf*PP1-null parasites

To refine PP1 function in the egress signaling cascade, we sought to bypass the egress phenotype of *Pf*PP1-null parasites using known inducers of the egress cascade: (i) the calcium ionophore A23187, a well-described inducer of egress, motility, microneme- and exoneme secretion and invasion [27,37–40] and (ii) the PDEs inhibitors Zaprinast and BIPPO that promote elevated levels of cGMP required to activate the downstream cGMP-dependent kinase PKG, essential to trigger merozoites egress [14,41–43]. To compare the effect of the molecules in a quantitative manner, we generated parasites expressing on an episome a nanoluciferase reporter in the PV lumen of *Pf*PP1-iKO parasites, hereafter referred to as *Pf*PP1-iKO-KnL line (Fig 1A, S1A Fig). For this, we fused the signal peptide of KAHRP protein to the nanoluciferase (KnL) to promote its secretion into the PV [28]. Upon merozoites egress, KnL is released in the culture supernatant and its activity can be quantified as a proxy for egress. Tightly

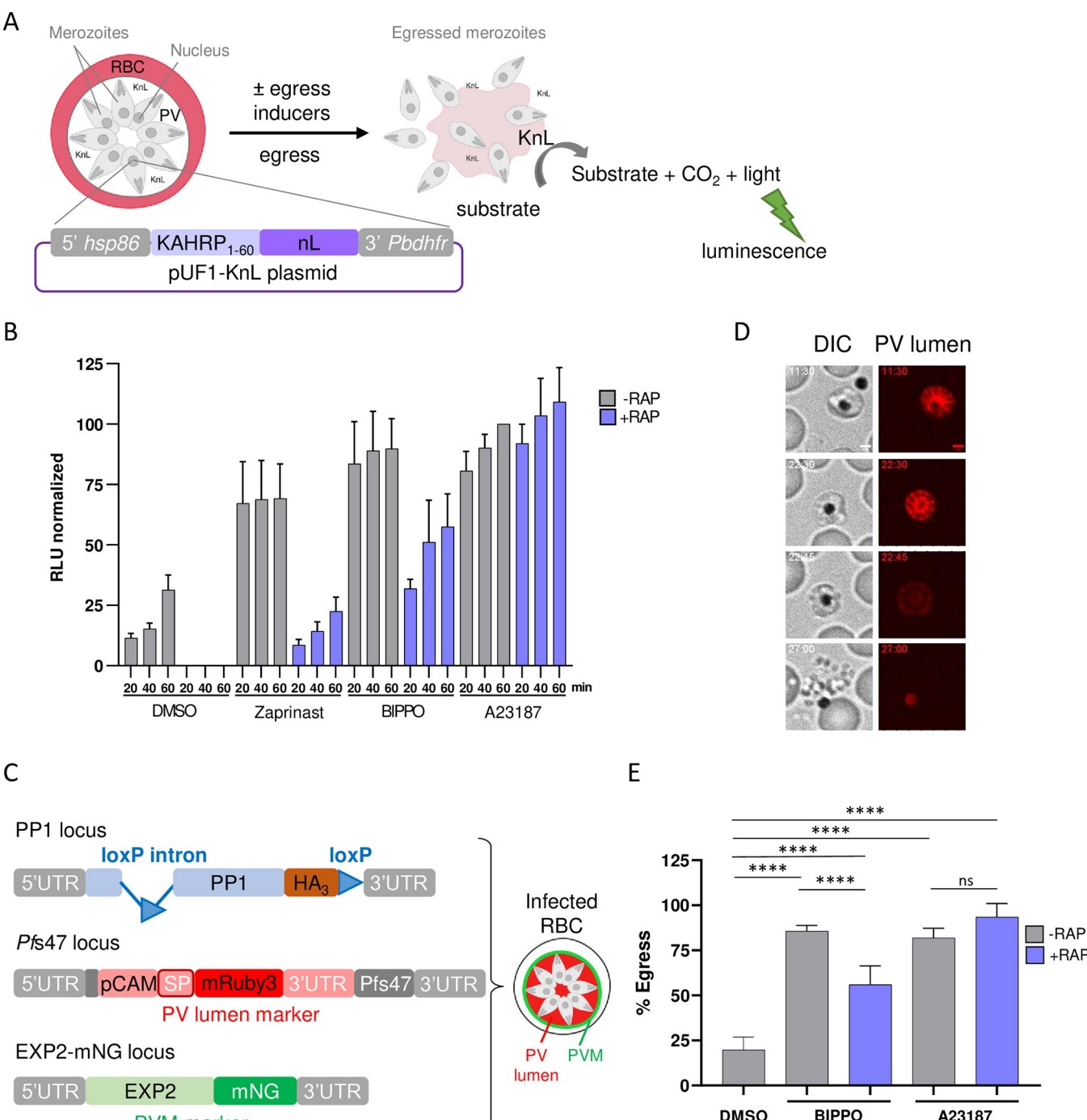

**Fig 1. Chemical genetic rescue of the egress block of *Pf*PP1-null parasites. A.** Schematic of KnL-based assay. *Pf*PP1-iKO-KnL parasites (Merozoites, light grey; nucleus, dark grey) harbor pUF1-KnL episomes, with KnL being secreted in the PV. The red cell represents the red blood cell (RBC). Upon egress, the enzyme is released in the extracellular medium, allowing to measure luminescence as a proxy for egress. The schematic was created with BioRender.com. **B.** Time-course of natural or induced egress of *Pf*PP1-iKO-KnL line. The luminescence (RLU) is expressed relative to the one measured at 60 min with control parasites (-RAP) induced with the calcium ionophore A23187. *Pf*PP1-iKO-KnL parasites treated ± RAP at 30 hpi were supplemented with vehicle (DMSO), 75 μM Zaprinast, 2 μM BIPPO or 10 μM A23187. Culture supernatants were collected at 20, 40 or 60 min post-treatment (n = 3 independent experiments). **C.** Schematic of the edited genomic loci of *Pf*PP1-iKOdt line. *pp1* gene is floxed to allow gene excision upon RAP treatment, and its coding sequence is in frame with a triple hemagglutinin tag (HA₃), as reported previously [34]. The PV lumen reporter mRuby3 fused to a signal peptide (SP) and under the calmodulin promoter (pCAM) has been inserted in the non-essential *Pf*s47 locus, while the PVM protein EXP2 has been tagged at the C-terminus with mNeonGreen (mNG). **D.** Still frames of a representative egress event of *Pf*PP1-iKOdt by live-microscopy. The numbers in the top left corner of each frame refer to time (min: sec). Scale bar: 2 μm. **E.** Quantification of egress by live-microscopy. Successful egress events of *Pf*PP1-iKOdt parasites were quantified among the segmented

parasites in the culture. Microscopic observations were repeated independently 5 times for DMSO and BIPPO, and 3 times for A23187. The total number of segmented parasites analyzed per condition: -RAP, DMSO n = 268; -RAP, BIPPO n = 222; -RAP, A23187 n = 77; +RAP, BIPPO n = 466; +RAP, A23187 n = 171. Statistical analyses were performed by one way Anova with Tukey's multiple comparisons test, with **** p ≤ 0.0001; ns = non-significant.

synchronized parasites were treated ± RAP at 30 hpi. Gene excision mediated by Cre recombinase at this timepoint leads to late PP1 depletion and blocking of egress only [34]. Mature schizonts of both cultures were collected by Percoll purification at 43 hpi, around the time of egress, as in our culture conditions the length of the erythrocytic cycle is ∼ 44 h. The time course of natural or induced egress was determined after the addition of vehicle or pharmacological treatment, respectively. The results were normalized to the luminescence measured during induced egress of control parasites (-RAP) at 60 min after A23187 treatment (Fig 1B). Natural egress (-RAP, DMSO) increased over time, reaching 31.4 ± 6.1% (mean ± SD) after only 60 min, reflecting a good enrichment of mature schizonts in our culture conditions. As expected, Zaprinast, BIPPO and A23187 promoted egress of most of the parasites within 20 min, thereby confirming their potent activity as egress inducers. Natural egress was completely blocked in PfPP1-null parasites (+RAP, DMSO), as reported previously [34]. However, this defect could be overcome by the 3 molecules, albeit at different rates. A23187 was the most potent molecule, causing a complete reversion of the egress block as early as 20 min, while Zaprinast and BIPPO induced a partial and progressive rescue, promoting up to 22.6 ± 5.8% and 57.6 ± 13.5% of egress respectively after 60 min.

To confirm these results at the single-cell level, we opted for time-lapse fluorescence videomicroscopy. For this, we modified the genome of PfPP1-iKO line in order to express two fluorescent reporters of the PV morphology, as described by Glushakova et al. [11]. The fluorophore mRuby3 was fused to a signal peptide to promote its secretion in the PV (Fig 1C). The construct, inserted in the non-essential Pfs47 locus [44], allows us to clearly identify segmented schizonts by live-microscopy due to their characteristic honeycomb pattern (Fig 1D, S2 Fig). The PVM protein Exp2 was also endogenously tagged with mNeongreen (mNG) to follow PV rounding-up and PVM rupture, and was used later in the study. This strain, referred to as PfPP1-iKOdt for "double tagged" was verified by PCR genotyping, sequencing and live microscopy (S2 Fig). Tightly synchronized PfPP1-iKOdt parasites treated ± RAP were collected around the time of egress and processed for live-microscopy. Guided by the mRuby3 honeycomb fluorescence pattern of segmented schizonts, we exclusively focused our analyses on this stage. We quantified the number of those that successfully egressed during the 30 min videos, either naturally, or following pharmacological treatment. Based on the fact that BIPPO had been more effective than Zaprinast at rescuing the egress defect of PfPP1-null parasites (Fig 1B), we restrained our study to BIPPO and A23187. Natural egress of control parasites (-RAP, DMSO; Movie A in S1 Video) represented 19.8 ± 7.1% (mean ± SD) of the segmented schizonts, and increased up to 85.7 ± 3.1% upon inducer treatment, with both molecules (Fig 1D and 1E, Movies B-C in S1 Video). Similarly, egress of stalled RAP-treated schizonts was also triggered by egress inducers, again at different rates, representing 56.0 ± 10.5% and 93.5 ± 7.5% of the parasites with BIPPO and A23187, respectively (Fig 1E, Movies D-E in S1 Video). These data are very concordant with the ones obtained using the KnL reporter, and together, show that PP1 depletion can be compensated by pharmacological inducers of the egress cascade. However, $Ca^{2+}$ mobilization induced by a calcium ionophore provides a better complementation to the egress defect of PfPP1-null parasites than cGMP increase by BIPPO, suggesting that the primary defect associated with PP1 depletion may be correlated to an inadequate $Ca^{2+}$ signaling.

Next, we investigated whether PfPP1-null merozoites that had been forced to egress by pharmacological treatment were viable and invasive. To do this, Percoll-purified schizonts

treated ± RAP at 30 hpi were supplemented with egress inducers at 43 hpi for 1h. Parasite multiplication rates were quantified by microscopic examination of blood smears. In control parasites (-RAP), we considered the multiplication rate resulting from natural egress (DMSO) to reflect the full invasive capacity of the parasites (S3 Fig). In contrast, BIPPO treatment led to the release of poorly invasive merozoites, consistent with the induction of premature egress, as previously reported for Zaprinast [14]. Similarly, treatment with A23187 prevented re-invasion of merozoites, which may also reflect premature egress and/or irreversible modification of uninfected RBCs by the calcium ionophore. *Pf*PP1-null parasites (+RAP) were blocked prior to the invasion step in DMSO-treated culture (S3 Fig). In contrast, BIPPO treatment led to productive invasion, with a multiplication rate consistently higher than that of control parasites treated with the same inducer (-RAP, BIPPO), though not statistically significant. We interpret this data as the result of the accumulation of egress-competent schizonts in the +RAP culture, which are therefore not prematurely activated by PDEs inhibition. Yet, the multiplication rate remained lower than the one observed during natural egress (-RAP, DMSO), indicating that some *Pf*PP1-null schizonts were prematurely activated. Finally, our results show that A23187, despite its ability to compensate for the egress defect, does not subsequently allow productive invasion.

## $Ca^{2+}$ mobilization by an ionophore restores normal kinetics of induced egress in *Pf*PP1-null parasites

To gain better insight in the kinetics of egress, we took advantage of the fluorescent PV reporters expressed in *Pf*PP1-iKOdt, and analyzed the morphological modifications of the vacuole by time-lapse video-microscopy (Fig 2A, Movie A in S2 Video). Initially, segmented schizonts displayed an irregular PV shape characterized by the asymmetrical Exp2-mNG labeling. Later, this staining showed a progressive redistribution towards the periphery of the parasite, indicative of an increase in vacuole sphericity and referred to as the rounding-up step. This step was shortly followed by PVM rupture, illustrated by a rapid decrease in the fluorescence intensities of both reporters, with mRuby3 being diluted in the host cytoplasm. Finally, the PVM was progressively dismantled and merozoite egress from the RBC coinciding with a complete loss of mRuby3 fluorescence.

As *Pf*PP1-iKOdt parasites allow to precisely follow the different steps of egress, we undertook a thorough analysis of their kinetics during natural and induced egress. To do this, tightly synchronized parasite cultures were harvested between 43 and 45 hpi to enrich for mature schizonts, and recorded 2 min. after pharmacological treatment, thus defining the start of kinetic measurements. We defined T1 as the duration of the entire egress process (Fig 2A). T2 corresponds to the time elapsed between the segmented schizont stage and the rupture of the PVM, the latter corresponding to the first frame with a single breach in the Exp2-mNG signal and concomitant with a sharp decrease in mRuby3 fluorescence. Finally, T3 corresponds to the time elapsed between the rupture of the PVM and the release of the merozoites, the latter event being evidenced by DIC and associated with a complete loss of mRuby3 signal. Natural egress (-RAP, DMSO) took on average 1177 ± 65 sec (~19 min) (mean ± sem), that can be divided into 807 ± 66 sec for T2 (~13 min) and 369 ± 27 sec for T3 (~6 min) (Fig 2B–2D). These values are in good agreement with the described morphological and molecular changes of the infected RBC associated with egress [9–11]. Of note, the distribution of the measured kinetics during natural egress was rather dispersed, suggesting that each segmented parasite was at a different stage of the egress pathway. In comparison, the overall duration of induced egress was significantly shorter (BIPPO: 659 ± 22 sec, ~11 min and A23187: 746 ± 30 sec, ~12 min), with T2 contributing predominantly to this acceleration (BIPPO: 368 ± 17 sec, ~6 min

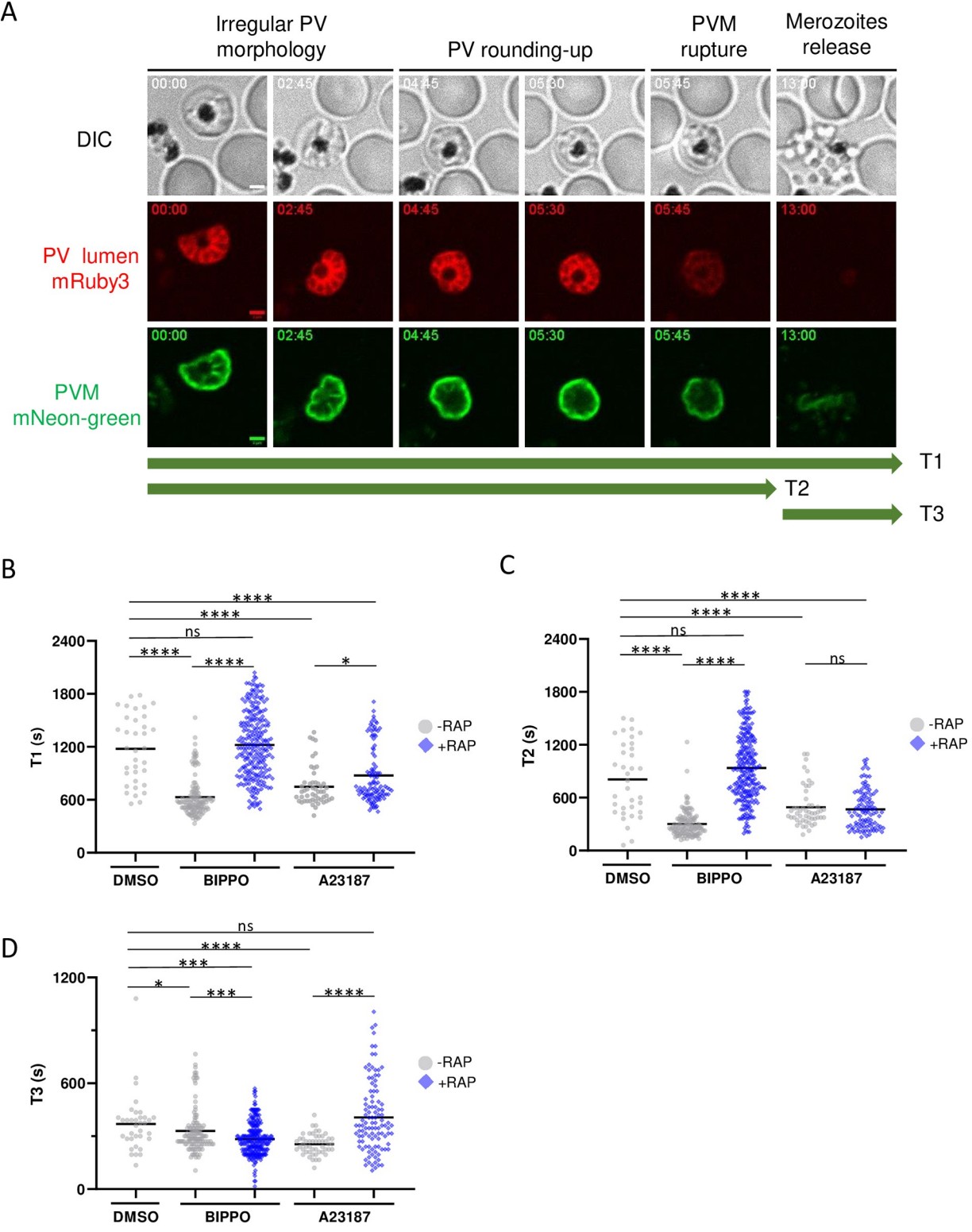

**Fig 2. Kinetics of natural and induced egress. A**. Still frames of a representative egress event of *Pf*PP1-iKOdt by live-microscopy. PV lumen and PVM correspond to the secreted mRuby3 or EXP2-mNG fluorescence, respectively. The different steps of egress are indicated on the top, and also correspond to the green arrows annotated T1 to T3, used to measure their kinetics throughout the study. The numbers in the top left corner of each frame refer to time (min:sec). Scale bar: 2 μm. **B-D**. Kinetics of *Pf*PP1-iKOdt egress in seconds (s). *Pf*PP1-iKOdt parasites treated ± RAP were supplemented with vehicle (DMSO), 2 μM BIPPO or 10 μM A23187. T1 to T3 represent the duration of the whole egress

process (B), to reach PVM rupture (C) or merozoites release (D), respectively. The solid bar represents the mean. Microscopic observations were repeated independently 5 times for DMSO and BIPPO, and 3 times for A23187. Number of parasites analyzed: -RAP, DMSO n = 35; -RAP, BIPPO n = 126; -RAP, A23187 n = 51; +RAP, BIPPO n = 252; +RAP, A23187 n = 108. Statistical analyses were performed with Mann-Whitney with **** $p \leq 0.0001$; *** $p \leq 0.005$; ** $p \leq 0.01$; * $p \leq 0.05$; ns = non-significant.

and A23187: 492 ± 31 sec, ~8 min) compared to T3 (Fig 2C and 2D, Movies B-C in S2 Video), indicating that egress molecules primarily modulate the kinetics of PVM breaching, and to a lesser extent merozoites release, the later may be explained by an enhanced secretion of egress-organelles. We also noticed that the distribution of the kinetics of induced egress was fairly homogenous, suggesting that BIPPO and A23187 likely trigger the egress pathway at a specific step. Overall, our data show that induced egress is faster than natural egress, and that PVM rupture is the step that is substantially accelerated.

We then compared the kinetics of rescued egress in *Pf*PP1-null parasites (+RAP) using the same inducers. Similar to control parasites, A23187 induced an overall decrease of the duration of egress (T1: 874 ± 31 sec; ~14 min.), correlated with an acceleration of T2 (T2: 468 ± 21 sec, ~8 min) as compared to natural egress (Fig 2B and 2C, Movie E in S2 Video). These results indicate that at the kinetic level, $Ca^{2+}$ mobilization overcomes the defect in PVM rupture of *Pf*PP1-depleted parasites. In sharp contrast, the kinetics of BIPPO-induced egress were fairly similar to the ones observed upon natural egress (T1: 1220 ± 23 sec, ~20 min; T2: 937 ± 24 sec, ~15 min) (Fig 2B and 2C, Movie D in S2 Video), indicating that in the context of PP1 depletion, BIPPO fails at promoting PVM rupture. In addition, the kinetics of merozoites release (T3) were disturbed with both inducers (Fig 2D), as compared to induced egress of control parasites, suggesting that PP1 may also play a role beyond PVM rupture. Altogether, the analyses of egress kinetics further support the view that PP1 depletion may result in a strong impairment in $Ca^{2+}$ mobilization, a phenotype that can only be bypassed and accelerated by a calcium ionophore.

## The alteration of cGMP homeostasis impacts the downstream $Ca^{2+}$ signaling in *Pf*PP1-null parasites

To shed light as to why BIPPO displayed a partial rescue capacity on *Pf*PP1-null parasites, we scrutinized the cGMP-PKG pathway in more detail. We had previously shown that PP1 depletion correlates with GCα hyperphosphorylation and lower parasitic cGMP levels [34]. To address whether the inhibition of PDEs by BIPPO is sufficient to reach threshold cGMP levels necessary to activate PKG, we compared by ELISA the cGMP levels of *Pf*PP1-iKO parasites treated ± RAP and collected around the time of egress. Percoll-purified schizonts (1,25.10$^8$ parasites per condition) were sampled 3 min following BIPPO or vehicle treatment. As expected for control parasites (-RAP), PDEs inhibition led to a robust and significant increase in cGMP levels, as compared to mock treatment (Fig 3A). As previously reported [34], *Pf*PP1-null schizonts exhibited lower cGMP concentrations but strikingly, this defect could not be fully compensated by PDEs inhibition. Despite a significant rise in cGMP levels in BIPPO condition, they remained lower than in control parasites (+RAP, BIPPO vs -RAP, BIPPO) (Fig 3A). Therefore, this result reveals a long-lasting impairment of GCα basal activity in *Pf*PP1-null schizonts.

As cGMP-dependent activation of PKG is linked to downstream $Ca^{2+}$ mobilization from intracellular stores [22], we assessed whether the aforementioned deficiency in cGMP levels of *Pf*PP1-null parasites may impact the downstream PKG-dependent $Ca^{2+}$ signaling. To evidence this cytosolic $Ca^{2+}$ signal, we used a genetically-encoded calcium indicator, GCamp6f [45], expressed from an episome under the pCRT promoter [46,47]. Following plasmid transfection

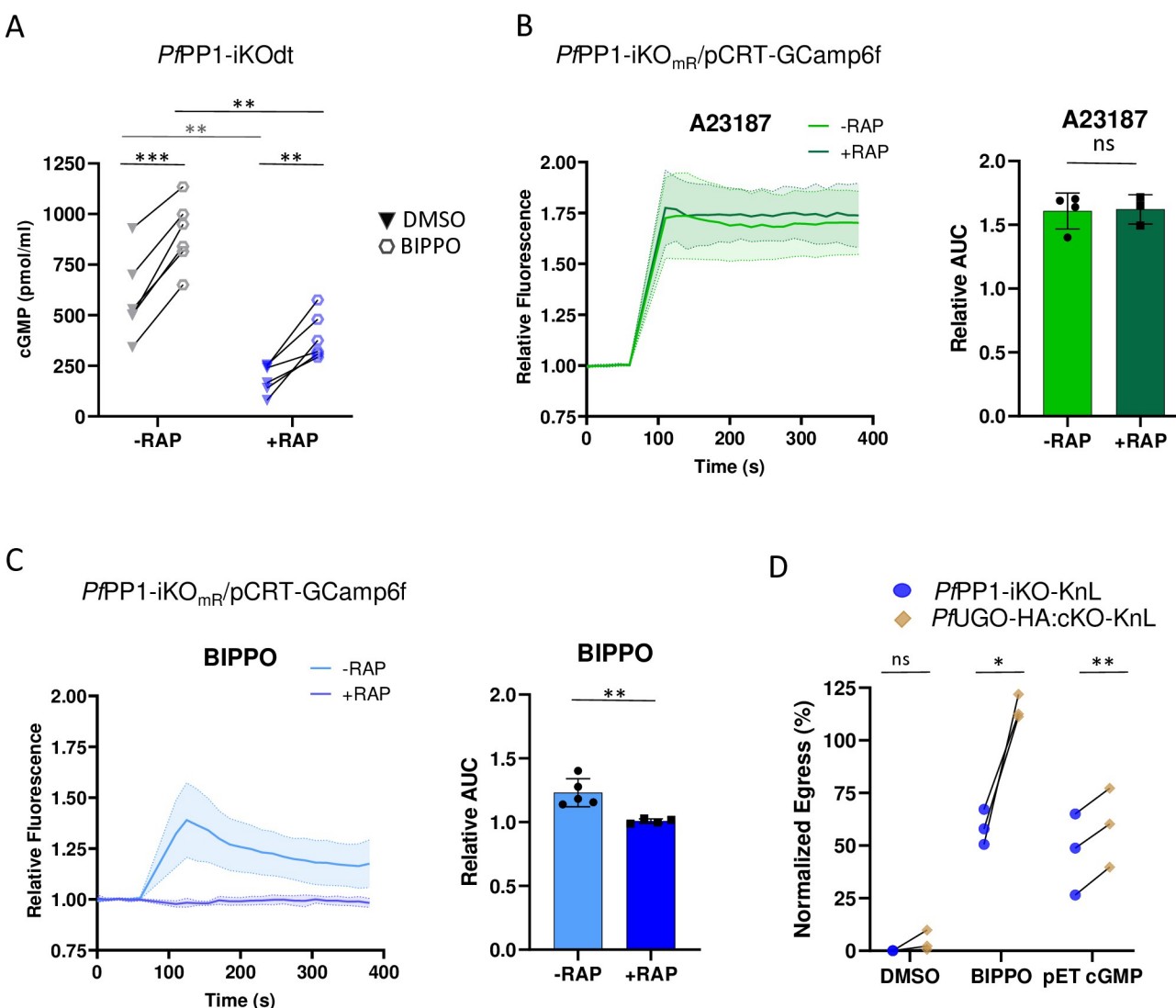

**Fig 3.** *Pf*PP1-null parasites are impaired in cGMP homeostasis and downstream PKG-dependent Ca2+ mobilization. **A**. Determination of cGMP levels in *Pf*PP1-iKOdt parasites treated ± RAP and supplemented with vehicle or 2 μM BIPPO. Statistical analyses were performed by a paired t-test with *** $p \leq 0.001$; ** $p \leq 0.005$ (n = 6 independent experiments). **B-C**. Determination of $Ca^{2+}$ levels in *Pf*PP1-iKO_{mR}/pCRT-GCamp6f strain in a saline buffer containing $Ca^{2+}$. Left: the fluorescence of GCamp6f was normalized to the baseline prior to the addition of the tested compound. Right: Comparison of the areas under the curve (AUC). Statistical analyses were performed by unpaired t-test with ** $p \leq 0.01$; ns = non-significant (n = 3–5 independent experiments). **D**. KnL-based egress assay. RAP-treated schizonts of *Pf*PP1-iKO-KnL or *Pf*UGO-HA:cKO-KnL were purified on Percoll and supplemented with vehicle, 10 μM A23187, 2 μM BIPPO, or 60 μM pET-cGMP. The results were normalized to the luminescence measured with A23187. Statistical analyses were performed by paired t-test with ** $p \leq 0.005$; * $p \leq 0.05$; ns = non-significant (n = 3 independent experiments).

in a *Pf*PP1-iKO line expressing mRuby3 in the PV lumen (*Pf*PP1-iKO_{mR}) (S4A and S4B Fig), GCamp6f fluorescence was tested by live-microscopy upon treatment with the calcium ionophore A23187. The expression of pCRT-driven GCamp6f led to a consistent increase in fluorescence upon A23187 treatment, despite a variable fluorescence background in each parasite likely due to copy number variations (S4C and S4D Fig). We then measured $Ca^{2+}$ mobilization of tightly synchronized *Pf*PP1-iKO_{mR}/pCRT-GCamp6f parasites treated ± RAP, and incubated with egress inducers or vehicle. A23187 and BIPPO both induced a specific rise in cytosolic $Ca^{2+}$ levels as compared to DMSO treatment (Fig 3B and 3C, S4E Fig), with a stronger and more long-lasting effect of the ionophore. Importantly, PP1-depletion (+RAP) completely

abolished BIPPO- but not A23187-induced $Ca^{2+}$ mobilization (Fig 3B and 3C). This difference may reflect distinct supplies of $Ca^{2+}$, with A23187 allowing $Ca^{2+}$ to cross any cellular membranes [48,49], while BIPPO would specifically trigger PKG-mediated $Ca^{2+}$ release from internal stores [22,26], thus consistent with a specific impairment in PKG activation. Yet, as we have shown earlier that BIPPO partially stimulates egress (Fig 1E), it is very likely that in some parasites, PDEs inhibition might be sufficient to stimulate the cGMP-PKG pathway, but the levels of PKG-dependent $Ca^{2+}$ release might be under the limit of our probe sensitivity or of its detection. To further verify the intracellular nature of the $Ca^{2+}$ stores, the experiment was repeated in a $Ca^{2+}$-depleted medium. A similar increase in GCamp6f fluorescence was observed with both molecules, the sole difference being that A23187 long-lasting effect was abolished (S4F Fig), thereby confirming that egress molecules-mediated $Ca^{2+}$ mobilization derives from intracellular stores, as already reported in *Plasmodium* and *T. gondii* [38,50]. Altogether, our results clearly establish a major defect of *Pf*PP1-null schizonts in cGMP homeostasis, and as a consequence, in PKG-dependent $Ca^{2+}$ mobilization.

Next, we reasoned that if the phenotype of *Pf*PP1-null parasites was solely linked to inadequate cGMP concentrations, it should be possible to overcome the egress defect by supplementing the cultures with pET-cGMP, a permeable analog of cGMP. Indeed, it was reported that the addition of this molecule reverted the egress defect of GCα-null schizonts [15]. To test this hypothesis, we used the *Pf*UGO-HA:cKO conditional mutant line as a control. UGO is a GCα-interacting protein, essential for natural egress, but not involved in GCα basal activity [19]. As its requirement can be bypassed by PDEs inhibition, we expected pET-cGMP to induce a similar reversion of the egress block. To compare the capacity of egress inducers at rescuing both strains, we expressed the KnL egress reporter in *Pf*UGO-HA:cKO line (S1B Fig), and tightly synchronized schizonts treated +RAP were Percoll-purified, before supplementing the cultures with vehicle, A23187, BIPPO or pET-cGMP. The results were normalized to the luminescence released upon A23187 treatment (Fig 3D). As previously reported [19], the egress phenotype of *Pf*UGO-null parasites was fully overcome by PDEs inhibition, unlike *Pf*PP1-null schizonts. Unexpectedly, pET-cGMP only induced a partial rescue of the egress block of *Pf*UGO-null schizonts, a result that may reflect the poor uptake of the molecule in our experimental conditions. Yet, this rescue was consistently higher than the one observed with *Pf*PP1-null parasites cultured in the same conditions, suggesting that *Pf*PP1-null schizonts might be affected in other processes than just cGMP homeostasis.

## PP1 regulates the calcium-dependent PV rounding-up step, prior to PVM rupture

Two early modifications of the PV have been described prior to PVM rupture, i.e. PV poration [8] and PV rounding-up [9–11], during which PP1 might also be involved in. We re-examined the egress videos of control *Pf*PP1-iKOdt parasites (-RAP) to evidence a possible leakage of mRuby3 in the host cytosol prior to PVM rupture, which would reflect PV poration. Out of 213 segmented schizonts analyzed, none displayed such a clear leakage and only one was considered having an ambiguous signal (Fig 4A), which prevented further analysis. PV poration was also evidenced by electron tomography and described as an equalization of contrast between the PV lumen and the host cell cytosol [8]. Therefore, we performed transmission electron microscopy on tightly synchronized *Pf*PP1-iKOdt parasites treated ± RAP at 30 hpi, and collected at the time of egress. To prevent natural egress of control parasites and enrich for schizonts with a porated PV as in Hale *et al.* [8], samples were treated with the PKG inhibitor Compound 2 (C2) [14,51]. In contrast to the published results [8], we could not observe an equalization of contrast between the two compartments, as 93–94 ± 6% (mean ± SD) of the

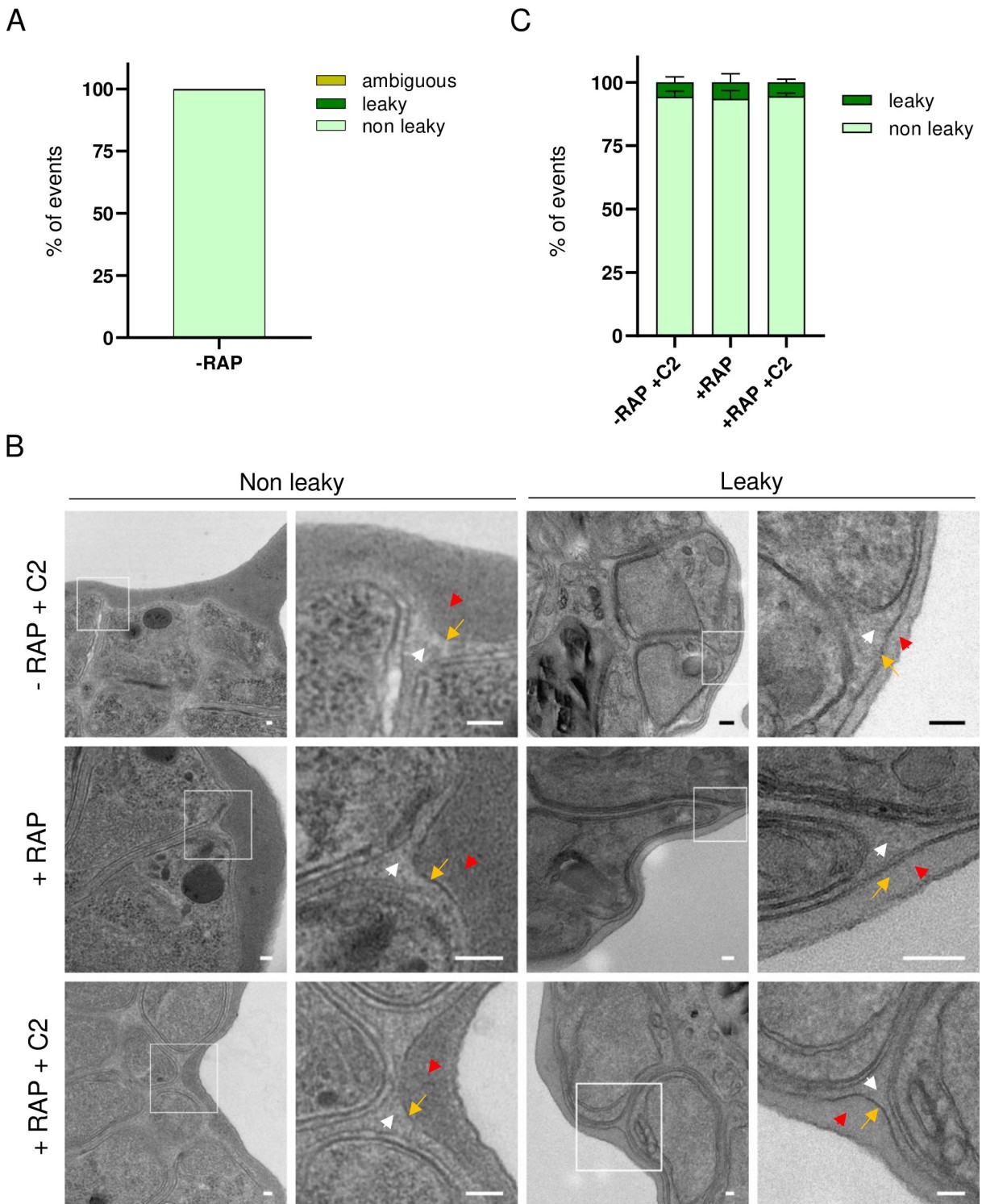

**Fig 4. Absence of evidence of PV poration. A**. Quantification of PV poration by live-microscopy. Leakage of mRuby3 fluorescence into the host cell cytosol was scrutinized for n = 213 segmented *Pf*PP1-iKOdt schizonts from n = 5 independent experiments. **B**. Representative electron microscopy images of *Pf*PP1-iKOdt treated ± RAP, blocked at 41 hpi with C2 and collected at 44 hpi. Most parasites displayed a PVM (yellow arrow), a clear difference of contrast between the PV lumen (white arrowhead) and the host cytosol (red arrowhead). Scale bar, 100 nm. **C**. Quantification of PV poration by TEM. The number of segmented schizonts analyzed per condition: -RAP + C2: n = 127; +RAP: n = 127; +RAP + C2: n = 112 (n = 2 experiments).

parasites displayed a clear contrast difference, irrelevant of the sample pre-treatment (Fig 4B and 4C). The reasons for the discrepancies with the results obtained by Hale *et al.* are not clear, and might be technical, but anyhow, PP1 depletion did not seem to modulate the PV poration step.

Next, we monitored PV rounding-up by time-lapse microscopy, a step likely $Ca^{2+}$-dependent as it can be inhibited by the calcium chelator BAPTA-AM, but independent of PKG as C2-treated schizonts arrest with a rounded vacuole [11]. Tightly synchronized *Pf*PP1-iKOdt were treated ± RAP at 30 hpi. Then, 2h before egress, the medium was supplemented with BAPTA-AM or C2, to prevent or enrich for rounding-up, respectively, or with vehicle. Samples were collected around the time of egress without Percoll purification to avoid any morphological modification of the infected cells prior to observation. We quantified the number of segmented schizonts that experienced PV rounding-up during the 30 min recordings. The change in PV morphology was observed in 42.3 ± 11.7% (mean ± SD) of control parasites (-RAP, DMSO) (Fig 5A, Movie A in S3 Video). As expected, this step was drastically inhibited by BAPTA-AM treatment, while C2 led to an enrichment of up to 59.6 ± 4.6% of parasites blocked with a more spherical vacuole (Movies B-C in S3 Video). Remarkably, the depletion of PP1 mimicked the effect of BAPTA-AM, with a strong inhibition of PV remodeling, thus demonstrating that PP1 is also involved in PV rounding-up (Movie D in S3 Video).

## $Ca^{2+}$ mobilization induces the rounding-up step and largely compensates for PP1-depletion

We wanted to determine whether the defect of *Pf*PP1-null parasites in PV rounding-up could be overcome by pharmacological treatment with egress inducers. For this, we restricted our analysis of the experiment described above to parasites that successfully egressed and quantified the number of those that displayed a peripheral redistribution of Exp2-mNG signal prior to PVM rupture. In our conditions, ~77.8 ± 22.5% (mean ± SD) of egressing control parasites (-RAP, DMSO) experienced the change in PV shape (Fig 5B), a percentage that we considered high enough to confirm that this morphological modification likely represents a normal feature of natural egress. A23187 or BIPPO treatment did not impact this percentage, indicating that induced egress as well follows a step-wise process that also begins with PV rounding-up prior to PVM rupture. Importantly, the defect in rounding-up of *Pf*PP1-null parasites (+RAP) was better overcome by A23187 than BIPPO, but remained partial in both conditions (Fig 5B). These results are in line with our previous observations on egress and also consistent with rounding-up being a $Ca^{2+}$-dependent mechanism.

To further analyze the kinetics of the rounding-up step, we discriminated the frames during which the parasites harbored an irregular PV morphology (T4) from the ones where the Exp2-mNG fluorescence started to relocalize towards the parasite periphery to give a more spherical pattern (T5) (Fig 5C). During natural egress (-RAP, DMSO), the mean duration of T4 was 743 ± 85 sec (~12 min) (mean ± sem), while PV rounding-up took on average 106 ± 17 sec (less than 2 min) (Fig 5D and 5E). Pharmacological treatment with egress molecules extensively shortened the duration of the irregular PV morphology T4 (A23187: 444 ± 51 sec, ~7 min; BIPPO: 350 ± 20 sec, ~5–6 min), without affecting the overall duration of rounding-up *per se* (T5). This reveals that the primary effect of egress inducers on the egress pathway is to promote the initiation of the rounding-up step. In *Pf*PP1-null schizonts (+RAP), this step was also stimulated by A23187, showing that $Ca^{2+}$ mobilization is necessary and sufficient to overcome the rounding-up phenotype. As anticipated, BIPPO failed at triggering this step in the mutant, a result that we attribute to its poor capacity to induce $Ca^{2+}$ release from internal stores. More intriguingly, the duration of the rounding-up *per se* (T5) was significantly

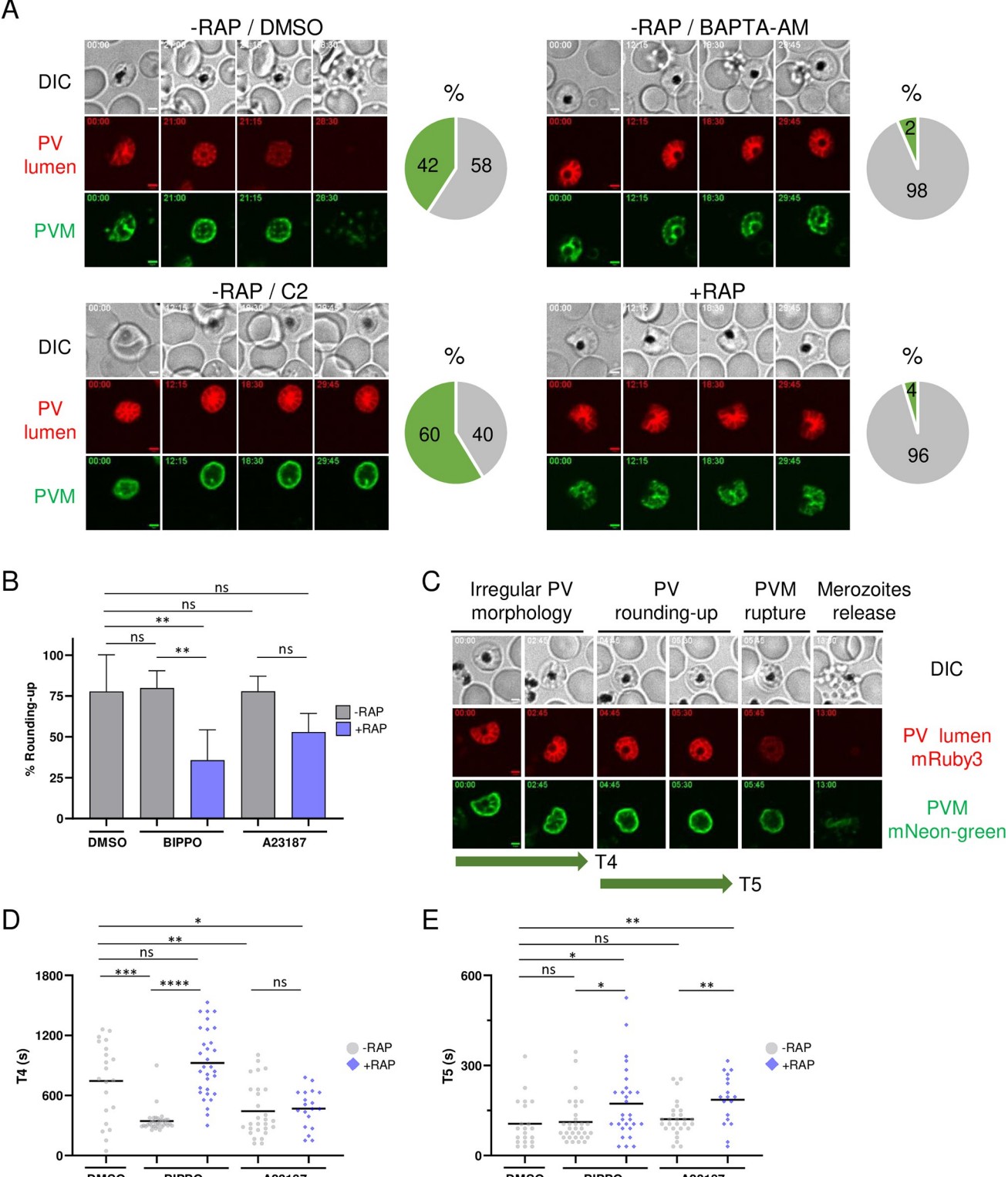

**Fig 5. *Pf*PP1-null parasites are impaired in the rounding-up step. A**. Quantification of PV rounding-up by live-microscopy. Left: Still frames of representative egress recordings of *Pf*PP1-iKOdt treated ± RAP. Control parasites (-RAP) were treated with vehicle, 60 μM BAPTA-AM or 1,5 μM C2. The numbers in the top left corner of each frame refer to time (min:sec). Scale bar: 2 μm. Right: pie charts representing for each condition the percentage of segmented schizonts that experienced PV rounding-up (green) or not (grey). The number of segmented schizonts analyzed per condition: -RAP, DMSO n = 262; -RAP, BAPTA-AM n = 124; -RAP, C2 n = 230; +RAP n = 241. (n = 5 independent experiments for DMSO, BAPTA-AM and C2 and n = 4

independent experiments for +RAP). **B**. Rescue of the PV rounding-up defect of *Pf*PP1-iKOdt line. *Pf*PP1-iKOdt parasites treated ± RAP were supplemented with vehicle, 2 µM BIPPO or 10 µM A23187 and recorded by live-microscopy. The percentage represents the number of egressing parasites that experienced rounding-up during the process. The number of parasites analyzed per condition: -RAP, DMSO n = 40; -RAP, BIPPO n = 127; -RAP, A23187 n = 52; +RAP, BIPPO n = 250; +RAP, A23187 n = 108. Statistical analyses were performed by unpaired t-test with ** $p \leq 0.005$; ns = non-significant (n = 5 independent experiments for DMSO and BIPPO and n = 3 for A23187). **C**. Still frames of a representative egress event of *Pf*PP1-iKOdt by live-microscopy, as in Fig 2A. The different steps of egress are indicated on the top, and also correspond to the green arrows annotated T4 and T5, used to measure their kinetics throughout the study. The numbers in the top left corner of each frame refer to time (min:sec). Scale bar: 2 µm. **D-E**. Kinetics of PV rounding-up in seconds (s). *Pf*PP1-iKOdt treated ± RAP were supplemented with vehicle (DMSO), 2 µM BIPPO or 10 µM A23187. T4 and T5 represent the irregular PV morphology (C), and PV rounding-up (D), respectively. The solid bar represents the mean. Number of parasites analyzed: -RAP, DMSO n = 21; -RAP, BIPPO n = 32; -RAP, A23187 n = 28; +RAP, BIPPO n = 29; +RAP, A23187 n = 18. Statistical analyses were performed by Mann-Whitney with **** $p \leq 0.0001$; *** $p \leq 0.005$; ** $p \leq 0.01$; * $p \leq 0.05$; ns = non-significant (n = 5 independent experiments for DMSO and BIPPO and n = 3 for A23187).

extended with both molecules, suggesting that PP1 may exert functions beyond $Ca^{2+}$ mobilization, or alternatively, that $Ca^{2+}$ release mediated by pharmacological treatment is not physiologically relevant to allow rounding-up to process normally in the absence of PP1. Altogether, our results demonstrate for the first time that pharmacological egress inducers stimulate the rounding-up of the PV and that PP1 phosphatase plays a critical regulatory function in this early step of the egress pathway of asexual schizonts.

## Discussion

Egress of blood-stage merozoites is a multi-step process exquisitely regulated. We are just starting to unveil its complexity, and despite a wealth of advances regarding the key players involved, many gaps remain in our understanding of its molecular pathway, especially regarding the early steps of the cascade. The identification of the parasite phosphatase PP1 as an essential enzyme that regulates organelle secretion and thus PVM rupture represented another hint pointing to the critical function of phospho-regulation during egress [34]. Among the proteins found to be hyperphosphorylated in PP1-depleted conditions was GCα, which provided a molecular link to the observed defect. Here, we generated a fluorescent reporter line allowing to monitor the different steps of egress by time-lapse video-microscopy to correlate morphological modifications of the PV with known effectors of the egress signaling pathway. Combined with chemical genetics using well-described pharmacological egress inducers, we could precisely (i) define the timing of natural and induced egress from hundreds of egress events, (ii) demonstrate the essentiality of PP1 for the rounding-up step and (iii) confirm the PP1-dependent regulation of GCα. Altogether, our work demonstrates the involvement of PP1 phosphatase in two distinct steps of the egress pathway and suggest a functional link between PP1 and $Ca^{2+}$ signaling.

Based on previously published PV reporters, we created the fluorescent *Pf*PP1-iKOdt line that could report for the described steps of egress, from PV rounding-up to PVM rupture and merozoite release. Using live-microscopy, we were able to document the kinetics of each step during natural egress. The PV of segmented schizonts displayed an irregular morphology for ~12 min, before its rounding-up during 1–2 min, shortly thereafter followed by PVM rupture (Fig 6). Then, it took on average ~6 min for the merozoites to breach the host cell. This egress kinetic is in good agreement with previous reports [11]. Additionally, we thoroughly analyzed induced egress triggered by A23187 or BIPPO. We found that these molecules accelerate the egress process, reducing its total duration by almost half. By promoting exocytosis of exonemes and micronemes, an event timely linked with PVM rupture [14,27,28], BIPPO and A23187 were expected to accelerate T2 (time to rupture the PVM). Yet, the acceleration of T3 (merozoite release from the RBC) was somewhat more surprising, and might reflect an enhanced discharge of secretory organelles upon pharmacological treatment. Such an effect may lead to an

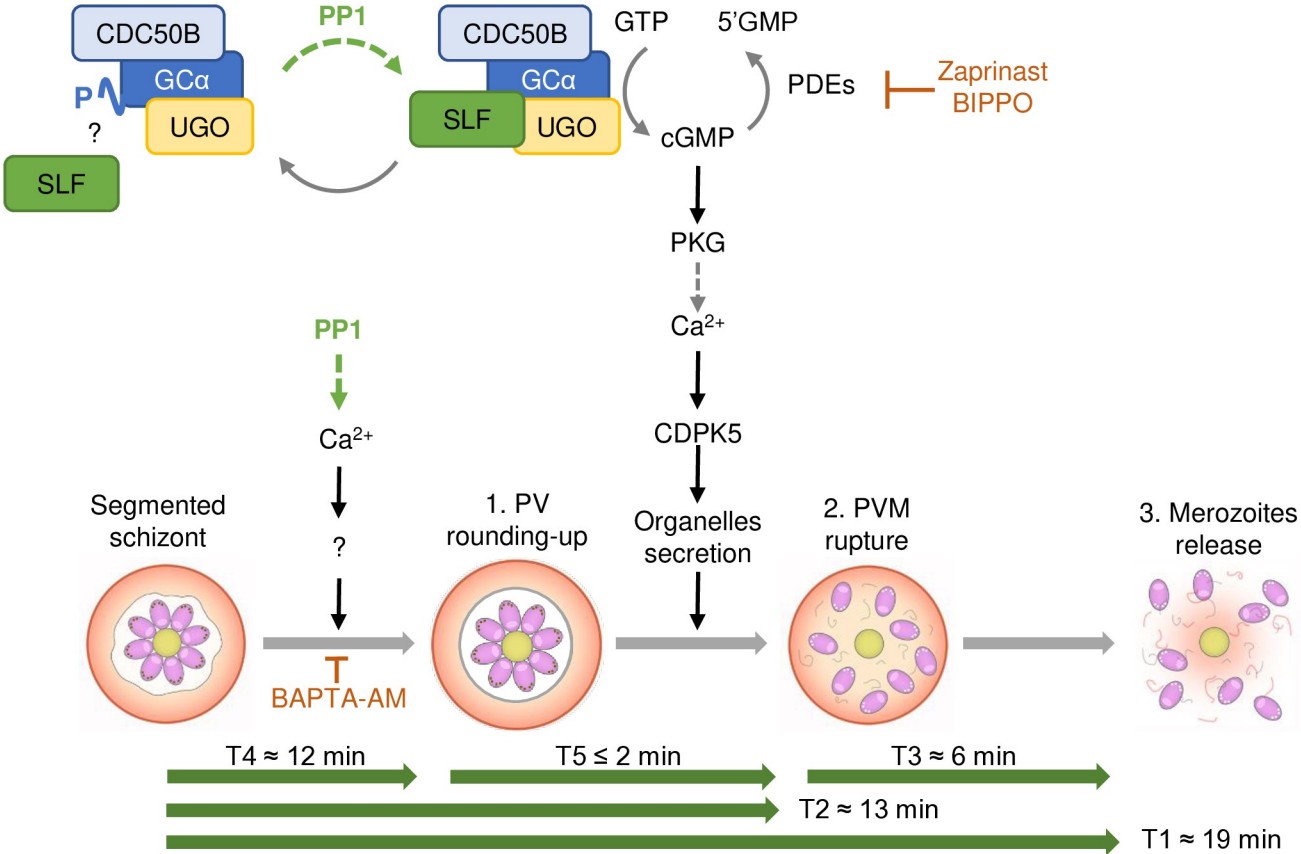

**Fig 6. Model of the asexual blood-stage egress pathway.** The different steps of egress and their kinetics (green arrows) determined in this study are depicted. On top, the molecular signaling pathway leading to egress recapitulates our current understanding of the cascade and the positioning of PP1 phosphatase deduced from this study and Paul et al. [34]. PP1 is involved in the initiation of the rounding-up step, possibly by controlling $Ca^{2+}$ release. In addition, PP1 also modulates cGMP homeostasis by regulating the phosphorylation status of GCα, which may adjust its interactions with cofactors like SLF. Dotted versus plain arrows represent indirect and direct mechanisms, respectively. Inhibitors of the egress pathway are shown in orange. SLF: Signaling Linking Factor; CDC50B: Cell Division Control 50B; UGO: Unique GC Organizer; GCα: Guanylyl Cyclase α; GTP: Guanosine Triphosphate; 5'GMP: Guanosine Monophosphate; PDEs: phosphodiesterases; cGMP: cyclic guanosine monophosphate; PKG: protein kinase G; CDPK5: calcium-dependent kinase 5.

accumulation of effectors in the PV such as SUB1, leading to enhanced processing/activation of its substrates, including SERA6 involved in the destabilization of the RBC cytoskeleton [29,30]. It is also possible that calpaïn 1, a calcium-regulated cysteine protease of the host erythrocyte might be activated more persistently upon treatment with egress inducers. As this host enzyme has been shown to play a role in *P. falciparum* egress, presumably by degrading erythrocyte cytoskeletal proteins, its sustained activation by $Ca^{2+}$ may also facilitate merozoites escape from the host cell [52]. Our results also reveal that those pharmacological agents promote PV rounding-up, rather than PVM rupture *per se*, an unexpected finding that fits well with rounding-up being a $Ca^{2+}$-dependent process [11]. Additionally, we could not evidence PV poration as an early step of egress in our conditions. In line with this observation, we show that natural and induced egress begin with the rounding-up step, supporting a model whereby an initial rise in intracellular $Ca^{2+}$ levels, yet to be characterized, would activate $Ca^{2+}$ effectors leading to PV rounding-up. The obligatory nature of this step suggests that it may be a pre-requisite to the sudden rupture of the PVM documented by video-microscopy. First reported as a swelling of the PV compartment correlated with the shrinkage of the erythrocyte [10],

volumetric analyses of the vacuoles refuted this interpretation and rather indicated the absence of increase in vacuolar volume associated with a decrease in the ratio between PV surface area and volume [11]. This membrane remodeling may induce modifications of its curvature and/ or tension, that may be necessary for phospholipase LCAT to trigger its collapse [33]. However, as this step of the egress cascade remains obscure, more work will be necessary to understand it at the mechanistic level.

Importantly, this study highlights a dual function for PP1 in the multi-step egress cascade. Firstly, we provide more evidence that PP1-dependent dephosphorylation of GCα modulates its enzymatic basal activity. It is to be noted that in *T. gondii*, GC was also found hyperphosphorylated in the absence of *Tg*PP1 [53]. We show that cGMP homeostasis is strongly affected in *Pf*PP1-null parasites, which translates into a defective PKG-dependent $Ca^{2+}$ mobilization. These phenotypes are not fully overcome neither by PDEs inhibition nor by pET-cGMP, and thus differ strikingly from other mutants defective in cGMP synthesis, including GCα-, CDC50B-, or UGO-null parasites [15,18,19]. The phenotype of *Pf*PP1-null parasites is more reminiscent of the depletion of SLF, another cofactor of GCα, that also shows a poor rescue by BIPPO, which led the authors to propose that SLF regulates GCα basal activity [19]. All the phosphorylation sites of GCα that are PP1-dependent are located in its P4-ATPase domain [34]. Although the precise interacting domains between GCα and its cofactors are unknown, PP1 may modulate the interface between GCα and SLF, or other partners, thereby regulating its basal activity (Fig 6), a hypothesis that remains to be validated. Secondly, we demonstrate that the phosphatase regulates PV rounding-up, a conclusion drawn from *Pf*PP1-null schizonts showing an almost complete inhibition of this step, and a disturbance of its kinetics even in the presence of egress inducers. To our knowledge, PP1 is the first regulator of PV rounding-up identified so far, which places the phosphatase upstream of the well-described cGMP-PKG node of the egress signaling pathway. Concordantly, C2 treatment impedes the egress cascade after PV remodeling, as shown previously [11]. Our data also support PV rounding-up to be a $Ca^{2+}$-dependent process, regulated by PP1. Indeed, the calcium ionophore A23187 overcomes the egress defect of *Pf*PP1-null schizonts, restores normal egress kinetics and triggers the initiation of rounding-up.

More importantly, the fact that PP1-depletion mimicked BAPTA-AM treatment suggests a strong connection between PP1 and $Ca^{2+}$ signaling (Fig 6). It has long been established that a complex interplay between kinases and phosphatases, especially PP1, regulate $Ca^{2+}$ homeostasis in mammalian cells, and its relevance was particularly studied in the context of myocytes and heart failure [54,55]. Several $Ca^{2+}$ transporters have been shown to be regulated in a PP1-dependent manner, including ryanodine and $IP_3$ receptors, the main $Ca^{2+}$ release channels of the sarcoplasmic reticulum, or the sarcoplasmic reticulum calcium-ATPase SERCA responsible for the uptake of $Ca^{2+}$ from the cytosol into the ER. This regulation takes place in various ways, through direct interaction of the phosphatase to the transporter [56], by modulation of the counteracting kinase activity such as $Ca^{2+}$/calmodulin-dependent protein kinase II [57,58], or *via* the regulation of a regulator of the channel activity, like phospholamban in the case of SERCA [59,60]. In the related apicomplexan parasite *T. gondii*, Herneisen *et al.* also found a strong link between *Tg*PP1 and calcium [53]. *Tg*PP1 showed $Ca^{2+}$-dependent thermal stability, and sub-minute phosphoproteomics comparing Zaprinast treatment and *Tg*PP1 depletion highlighted global dysregulation of ion homeostasis, including $Ca^{2+}$ ATPases, which correlated with delayed $Ca^{2+}$ mobilization and reduced $Ca^{2+}$ entry and resting $Ca^{2+}$ concentrations [53]. Even though we did not identify the orthologs of those transporters as phosphorylated in a PP1-dependent manner in our phosphoproteomic dataset, likely due to low temporal resolution [34], several phosphorylation sites have been reported in SERCA [61–64], suggesting a plausible regulation by this post-translational modification. As our results also

suggest PP1-dependent $Ca^{2+}$ regulation, we speculate that in *Plasmodium*, PP1 might also control $Ca^{2+}$ entry and/or release from internal stores, responsible for PV rounding-up to take place, a hypothesis that will require further investigation.

In conclusion, this study sheds new light on the role of a highly conserved serine/threonine phosphatase in the regulation of the egress pathway of *Plasmodium* asexual blood-stage schizonts. As the modulation of PP1 complexes appears as a potential target for cancer treatment [65], this axis of research development could also benefit individuals affected by human malaria in the future.

## Materials and methods

### Ethics statement

Human blood from anonymous donors was used, and the approval number 21PLER2016-0103 was issued from the French blood bank.

### Parasite strains, cell culture and transfection

The inducible knock-out line *Pf*PP1-iKO previously generated [34] was used as the genetic background for further genetic modifications. It is to be noted that this line exhibits a 42-44h erythrocytic cycle in our culture conditions. The parasites were cultured in RPMI 1640 medium (Gibco) supplemented with 10% human serum and 40 μg/ml gentamycin in human erythrocytes from anonymous donors (French blood bank, approval number 21PLER2016-0103). The cultures were placed at 37°C in incubator chambers and maintained in hypoxic conditions (5% $O_2$, 5% $CO_2$, 90% $N_2$). Parasites were synchronized by enrichment of late schizonts on a 70% percoll cushion, followed 1–2 hours later by 5% D-sorbitol treatment. When required, mock (DMSO) or 20 nM Rapamycin was added post-reinvasion to induce *ugo* gene excision, or at 30 hours post-invasion (hpi) to induce late PP1 depletion, as shown in [34].

To transfect *P. falciparum* parasites, 60 μg of plasmid DNA were resuspended in cytomix (120 mM KCl, 0,15 mM $CaCl_2$, 2 mM EGTA, 5 mM $MgCl_2$, 10 mM $K_2HPO_4/KH_2PO_4$, pH 7,6, 25 mM Hepes/KOH, pH 7,6) and mixed with 100 μl of ring-stage parasites. Electroporation was performed using a BTX electroporator at 310 V, 950 μF. Selective drug pressure was applied 5h post-transfection. Following selection, parasites were cloned by limiting dilution, genotyped by PCR and the modified locus sequenced.

### Reagents and antibodies

All the following reagents were prepared in DMSO: Rapamycin (LC laboratories), Zaprinast (Sigma), A23187 (Sigma), BIPPO [42], C2 [51] (gift from Dr O. Bilker), DSM1 (Calbiochem), WR99210 (Jacobus Pharmaceuticals), BAPTA-AM (Sigma). Blasticidin (Sigma) was resuspended in RPMI 1640. pUF1-KnL was selected with 1 μg/ml DSM1; pLN-mRuby3 and pLN-PfEXP2-mNG were derived from pLN-ENR-GFP [66] and selected with 2,5 μg/ml Blasticidin. pCRT-G6f was derived from pARL2-GFP [67] and selected with 2,5 nM WR99210; pDC2-gRNA plasmids encoding for the gRNA and the Cas9 nuclease were selected with 2,5 nM WR99210.

### Molecular biology

All the primers used in this study are listed in S1 Table. PCRs were performed with Q5 DNA polymerase (NEB) and amplicons verified by sequencing.

pLN-mRuby3. To generate mRuby3 as a PV lumen reporter, we fused the signal peptide of HSP101 (first 27 amino acids; $SP_{HSP101}$) to mRuby3 coding sequence. To do so, we amplified separately $SP_{HSP101}$ from *Pf*3D7 genomic DNA with primers MLa375/MLa376 and mRuby3 with primers MLa377/MLa378 from pBSL30-Rab7a [68]. The gene fusion, was obtained by mixing both amplicons and the primers MLa375/MLa378, and cloned AvrII/AflII in pLN-ENR-GFP so as to be expressed under the calmodulin promoter [66]. We chose the *Pfs47* locus to introduce the mRuby3 cassette, as it is non-essential during the parasite blood-stage asexual development [44]. To allow for recombination to take place, we cloned two homology regions (HR) on both sides of the cassette. HR1 and HR2 were amplified from *Pf*3D7 using primers MLa371/MLa372 and MLa373/MLa374, and cloned ApaI/HpaI and PstI/BamHI, respectively.

pLN-*Pf*Exp2-mNeonGreen (pLN-*Pf*EXP2-mNG). This plasmid was generated by replacing the GFP coding sequence of pLN-*Pf*EXP2-GFP [34] by the mNeonGreen fluorescent protein [69]. To do so, the mNeonGreen coding sequence flanked in 5' by a glycine-serine linker (5 repetitions GS) was amplified using primers MLa369/MLa370 from pPLOT-mNG plasmid [70] and cloned BsiWI/AflII in the pLN-*Pf*EXP2-GFP in place of the *gfp* coding region.

pUF1-Kharp-Flag-Nanoluciferase (pUF1-KnL). To express the nanoluciferase in the PV, we fused the first 60 amino acids of KAHRP to the coding sequence of the nanoluciferase (KnL) as described in [28], as these 60 amino acids have been shown to be sufficient to drive expression of a GFP reporter into the PV lumen [71]. A nanoluciferase was amplified using primers MLa352/MLa353 from pCAGGS-NanoL [72], while the 5' of the *kahrp* gene was amplified with MLa350/MLa351 from *Pf*3D7. Mixing the 2 PCR products in the same reaction with primers MLa350/MLa353 led to the amplification of the 717 bp fragment corresponding to the KAHRP-nL (KnL). This fragment was cloned in place of the *cas9* gene in the pUF1-Cas9 plasmid [73] opened XhoI/KpnI.

pARL2-pCRT-GCamp6f (pARL2-G6f). The calcium sensor GCamp6f was amplified using primers MLa462/MLa464 from pGP-CMV-GCamp6f plasmid (Addgene) and cloned AvrII/XmaI in the pARL2-GFP vector [67], in place of the GFP coding sequence.

pDC2-gRNAs. The primers forward and reverse corresponding to the gRNAs (S1 Table) were hybridized in a PCR machine and ligated in the pDC2-cam-Cas9-U6-hDHFR plasmid [74] opened with BbsI enzyme. The gRNA insertion was verified by PCR screen and further validated by sequencing.

## Egress assay using the KnL reporter

Synchronous *Pf*PP1-iKO-KnL or *Pf*UGO-HA:cKO-KnL pre-treated ± RAP were Percoll-purified around 43 hpi, and parasitemia was adjusted around 1% in complete medium. Each culture was immediately sampled (T0) for background control and to verify the parasitemia of each culture. The remaining schizonts ($4.10^4$ parasites/condition) were supplemented with vehicle (DMSO), 75 μM Zaprinast, 2 μM BIPPO, 10 μM A23187 or 60 μM pET-cGMP and incubated at 37°C. Cultures were sampled at 20, 40 or 60 min for all the compounds, except for pET-cGMP, for which samples were collected after 2h. Following 5 min centrifugation at 1000 rpm (130g) to pellet the cells, 40 μl of supernatants were transferred in a luminometer 96 wells plate and processed with Nano-Glo Luciferase Assay Reagent according to the manufacturer's instructions (Promega N1110). The plate was read 3 min after the addition of the reagents on a Berthold luminometer. The T0 background was first subtracted from the results, then the luminescence was normalized to the parasitemia of each sample and to the luminescence measured for control parasites treated with A23187 for 60 min (100%).

## Invasion assays

Synchronous *Pf*PP1-iKOdt pre-treated ± RAP were Percoll-purified around 43 hpi, and parasitemia was adjusted around 2% in complete medium. Each culture in technical replicates was immediately sampled (T0) for microscopic examination, or supplemented with vehicle (DMSO), 2 μM BIPPO or 10 μM A23187 and incubated at 37˚C. After an incubation of an hour, cultures were washed two times in complete medium and smeared for further analysis. Thin blood smears were stained with Giemsa. Counts of schizonts and ring stages were performed blindly on 5.000 to 10.000 cells per blood smear by 2 researchers. The multiplication rate represents the ratio between the percentage of ring stages and the percentage of schizonts that egressed during the course of the experiment.

## Time-lapse video microscopy

Live microscopy was performed on tightly synchronised *Pf*PP1-iKOdt schizonts pre-treated ± RAP at 30 hpi. Images were acquired every 15 s for 30 min with a Spinning disc Nikon Ti Andor CSU-W1, using the following parameters: DIC: 50ms exposure; 488 TRIO 521–630 FAST: 100 ms exposure; 561 TRIO 521–630 FAST: 200 ms exposure. To test whether egress inducers could overcome the egress defect of *Pf*PP1-null parasites, 50 μl of mature schizonts at 0.3% haematocrit were sedimented for 10 min on a μ-slide VII0.4 Poly-L-lysine (Ibidi #81604), before being supplemented with vehicle, 10 μM A23187 or 2 μM BIPPO. The acquisition was started 2 min after the addition of the egress molecules. To determine whether PP1 was involved in the rounding-up step, *Pf*PP1-iKOdt parasites were treated ± RAP at 30 hpi, and mature schizonts were collected at ~43 hpi. Control parasites (-RAP) were treated with vehicle, 60 μM BAPTA-AM or 1.5 μM C2 for an hour and sedimented in a μ-slide 8 well glass bottom (Ibidi #80827) before microscopic acquisitions. To record $Ca^{2+}$ signalling in the parasite, 250 μl of *Pf*PP1-iKOmR/pCRT-Gcamp6f parasites at 0.1% haematocrit were settled for 30 min on a μ-slide 8 well glass bottom coated with Cell-Tak (Corning 354240), before the addition of 50 μl of 60 μM A23187. The parameters used were 488 TRIO 521–630 FAST: 200 ms exposure; 561 TRIO 521–630 FAST: 100 ms exposure. Images were collected every 2 s for 20 min.

The videos were analysed with FIJI. To determine the kinetics of egress, the morphological changes associated with egress were checked frame-by-frame for each parasite. To assess GCamp6f fluorescence, we used the Median Radius 3 filter and applied the Li threshold. The mean fluorescence intensity (MFI) was extracted for each frame and calculated by FIJI.

## Measurement of cGMP levels

cGMP levels were measured using Direct cGMP ELISA kit (Enzo Life Sciences) on *Pf*PP1-iKOdt parasites treated ± RAP at 30 hpi. $1,25.10^8$ Percoll-purified schizonts per condition were incubated for 3 min in complete medium supplemented with DMSO, 2 μM BIPPO or 10 μM A23187. Parasites were pelleted 1 min at 600g before being resuspended in 100 μl of 0.1M HCl and incubated 3 min at room temperature. After another centrifugation step for 5 min at 9000g, supernatants were collected, and diluted with 400 μl of 0.1M HCl, before freezing at -80˚C. cGMP levels were measured with acetylated protocol according to the manufacturer's instructions. Absorbance was detected at 405nm on a TECAN Microplate Reader Sunrise.

## Measurement of $Ca^{2+}$ mobilization

*Pf*PP1-iKO_mR/pCRT-GCamp6f parasites were tightly synchronized and treated ± RAP at 30 hpi. Percoll-purified schizonts were washed in a saline solution (154 mM NaCl; 4 mM KCl; 10

mM HEPES; 0.15 mM MgCl$_2$; 10 mM Glucose; 0.4 mM CaCl$_2$; pH7.2) or in a saline solution deprived of Ca$^{2+}$ (145 mM NaCl; 4 mM KCl; 10 mM HEPES; 0.15 mM MgCl$_2$; 10 mM Glucose; 100 µM EGTA; pH7.2). 100 µl of parasites at $7,5.10^7$ parasites/ml were added in a 96-wells plate. Each condition was done in technical duplicate. Baseline fluorescence was read in a microplate reader (TECAN Microlate Reader Sunrise) pre-warmed at 37˚C, at 15 s intervals for 1 min (excitation: 488 nm; emission: 533 nm) before the addition of egress inducers (10 µM A23187; 2 µM BIPPO). GCamp6f fluorescence was again read at 15 s intervals for another 5 min. The fluorescence was normalized to the baseline fluorescence.

## Transmission electron microscopy

Tightly synchronized *Pf*PP1-iKOdt parasites were treated ± RAP at 30 hpi and their egress from the host cell was blocked at 41 hpi with C2. Mature schizonts were collected at 44 hpi by Percoll purification and fixed by adding 25% glutaraldehyde EM grade directly in the culture medium in order to obtain a final concentration of 2.5%. After 10 min at RT, cells were centrifuged and the pellet resuspended in 20 pellet volume of cacodylate buffer 0.1M containing 2.5% glutaraldehyde for 2h at RT. Samples were then stored at 4˚C until subsequent processing. All the following incubation steps were performed in suspension, followed by centrifugation using a benchtop microcentrifuge. Incubation and washing steps were performed in Pelco Biowave PRO+ Microwave processing system (TED Pella). Program details are provided in S2 Table. Cells were washed with cacodylate buffer and post-fixed with 1% O$_s$O$_4$ immediately followed by 1.5% potassium ferricyanide in the same buffer. After washing with distilled water, samples were incubated in 2% uranyl acetate for 30 min at 37˚C and further processed in the microwave. Following another washing step, samples were incubated in lead aspartate pre-heated to 50˚C. Dehydration was performed with increasing concentrations of acetonitrile. Samples were then impregnated with EMbed-812 resin and polymerized for 48 hours at 60˚C. All chemicals were obtained from EMS. 70 nm thin sections were made using a UCT ultramicrotome (Leica) equipped with an Ultra 35˚ diamond knife (Diatome), and collected on 100-mesh Formvar coated grids. Sections were imaged on a transmission electron microscope LaB6 JEOL 1400 Flash at 100 kV at the Electron Microscopy facility of the University of Montpellier (MEA).

## Supporting information

**S1 Fig. Verification of the presence of pUF1-KnL plasmid by PCR.** PCR verification of the presence of the KnL reporter in *Pf*PP1-iKO-KnL (A) and *Pf*UGO-HA:cKO-KnL (B) parasite lines, as compared to the parental background. The gene encoding the lactate dehydrogenase (*ldh*) was used as a control. bp: base pairs.
(TIF)

**S2 Fig. Generation of *Pf*PP1-iKOdt fluorescent line. A**. Schematic of the C-terminal tagging of *exp2* with mNG by CRISPR-Cas9. Green rectangles and lines represent exons and introns, respectively. Dashed lines show the double homologous recombination taking place in the *exp2* locus. Red lightning stands for Cas9 double strand break. Integrative PCRs as in B are shown on the edited locus. **B**. PCR genotyping of *Pf*PP1-iKO/Exp2-mNG parasites, as compared to the parental line. kbp: kilo base pairs. **C**. Schematic of the integration of mRuby3 in the *Pf*s47 locus of *Pf*PP1-iKO/Exp2-mNG. Dashed lines show the double homologous recombination taking place at the *Pf*s47 locus. Red lightning stands for Cas9 double strand break. Integrative PCRs as in D are shown on the edited locus. **D**. PCR genotyping of *Pf*PP1-iKOdt parasites, as compared to the parental line. Amplification of the *Pf*s47 locus shows the

extension of the locus due to mRuby3 construct integration in the edited parasites. kbp: kilo base pairs. **E.** Validation of *Pf*PP1-iKOdt parasite line by live-microscopy. Scale bar = 2 μm.
(TIF)

**S3 Fig. Invasive capacities of merozoites from induced egress.** Quantification of invasion on Giemsa-stained blood smears. The multiplication rate upon egress of *Pf*PP1-iKOdt parasites treated ± RAP was quantified following 1 h of pharmacological treatment by microscopic examination of thin blood smears and counting of schizonts and ring stages (n = 3 biological replicates). Statistical analyses were performed by one way Anova with Tukey's multiple comparisons test, with ** $p \leq 0.005$; * $p \leq 0.05$; ns = non-significant.
(TIF)

**S4 Fig. Generation of a calcium reporter parasite line. A**. PCR genotyping of *Pf*PP1-iKO$_{mR}$ parasites, as compared to the parental line. **B**. Still frames of *Pf*PP1-iKO$_{mR}$ parasites from live microscopy. Scale bar = 2 μm. **C-D**. Top: Mean fluorescence intensities (MFI ± SD) of 5 representative parasites from *Pf*PP1-iKO$_{mR}$ line (C), or *Pf*PP1-iKO$_{mR}$-pCRT-GCamp6f line (D). MFI was normalized to the fluorescence baseline recorded prior to the addition of A23187. Bottom: Still frames from live-microscopy experiment performed on the same parasites as depicted above. **E.** Determination of Ca$^{2+}$ levels in *Pf*PP1-iKO$_{mR}$/pCRT-GCamp6f strain in a saline buffer containing Ca$^{2+}$, upon treatment with vehicle. Left: the fluorescence of GCamp6f was normalized to the baseline prior to the addition of DMSO. Right: Comparison of the areas under the curve (AUC). Statistical analyses were performed by unpaired t-test with ** $p \leq 0.01$; ns = non-significant (n = -5 independent experiments). **F.** Determination of Ca$^{2+}$ levels in *Pf*PP1-iKO$_{mR}$/pCRT-GCamp6f strain in a saline buffer deprived of Ca$^{2+}$. The fluorescence of GCamp6f was normalized to the baseline prior to the addition of the tested compound. n = 3 or 2 independent experiments for A23187 and BIPPO, respectively.
(TIF)

**S1 Video. Rescue of the egress block of *Pf*PP1-null parasites with egress inducers.** Egress movies of representative parasites pre-treated -RAP (A-C) or + RAP (D-E) during natural egress (A), or induced egress with BIPPO (B and D) or A23187 (C and E).
(ZIP)

**S2 Video. Kinetics of natural or induced egress of *Pf*PP1-iKOdt parasites.** Egress movies of representative parasites pre-treated -RAP (A-C) or + RAP (D-E) during natural egress (A), or induced egress with BIPPO (B and D) or A23187 (C and E).
(ZIP)

**S3 Video. PP1 depletion prevents the rounding-up step.** Egress movies of representative parasites pre-treated -RAP (A-C) or + RAP (D). In addition, control parasites (-RAP) were supplemented with BAPTA-AM (B) or C2 (C).
(ZIP)

**S1 Table. List of primers used in the study.**
(XLSX)

**S2 Table. Program details for TEM samples preparation.**
(XLSX)

**S3 Table. Raw values and analyses related to the main figures.**
(XLSX)

**S4 Table. Raw values and analyses related to the Supplemental figures shown in Appendix.** (XLSX)

## Acknowledgments

We thank the staff of the MRI-platform for imaging and the Electron Microscopy facility of the University of Montpellier for assistance and technical support. We thank Pr. Y. Sterkers, and Drs A. Gross, R. Gaudin for plasmids, Dr O. Bilker for C2 compound, and Pr. M. Brochet for sharing *Pf*UGO-HA:cKO parasite line. This work was supported by the Agence Nationale de la Recherche (ANR-21-CE15-0043), and the Laboratoire d'Excellence Parafrap (ANR-11-LABX-0024).

## Author Contributions

**Conceptualization:** Mauld H. Lamarque.

**Formal analysis:** Marie Seveno.

**Funding acquisition:** Maryse Lebrun, Catherine Lavazec, Mauld H. Lamarque.

**Investigation:** Marie Seveno, Manon N. Loubens, Laurence Berry, Arnault Graindorge.

**Supervision:** Mauld H. Lamarque.

**Validation:** Mauld H. Lamarque.

**Visualization:** Marie Seveno, Mauld H. Lamarque.

**Writing – original draft:** Mauld H. Lamarque.

**Writing – review & editing:** Maryse Lebrun, Catherine Lavazec, Mauld H. Lamarque.

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
