## [Decision Letter · Decision Letter 0]

25 Aug 2024

Dear Dr. Lamarque,

Thank you very much for submitting your manuscript "The malaria parasite PP1 phosphatase controls the initiation of the egress pathway of asexual blood-stages by regulating the rounding-up of the vacuole" for consideration at PLOS Pathogens. As with all papers reviewed by the journal, your manuscript was reviewed by members of the editorial board and by several independent reviewers. The reviewers appreciated the attention to an important topic. Based on the reviews, we are likely to accept this manuscript for publication, providing that you modify the manuscript according to the review recommendations.

Sincerely,

Christopher J. Tonkin

Guest Editor

PLOS Pathogens

Dominique Soldati-Favre

Section Editor

PLOS Pathogens

Michael Malim

Editor-in-Chief

PLOS Pathogens

orcid.org/0000-0002-7699-2064

Reviewer Comments (if any, and for reference):

Reviewer's Responses to Questions

**Part I - Summary**

Reviewer #1: The present manuscript by Saveno et al describe how the PP1 phosphatase is involved in the egress signalling mechanism during malaria parasite asexual blood stages. They address this by using their previously generated PP1 mutant in human malaria parasite P. falciparum that they published earlier by the same group. For this they utilised chemical genetics using various ionophore and Ca inducer and live cell imaging. Their result demonstrate that PPi1has dual role during this signalling cascade of egree of parasite from the red cell. They show that one function is that PP1 regulate the basal activity of guanyl cyclase and second is that PP1 controls the rounding of the parasitophororus vacuole(PV) that essential fo the egress of the parasite.

They very elegantly showed that calcium ionophore A23187 can bypass this egress defect of PP1null mutant and can initiate the rounding up of the vacuole. This is the most important finding of the paper. They further tease out this Ca signalling with Zaprinast, BIPPO and propose that cGMP-PKG as target of the phosphatase.

The manuscript is well written and most experiments well with their aims. The authors present interesting finding and there are very few studies on the phosphatase signalling and especially PP1 in parasites and hence it advances the field. There few minor point that could be easily addressed.

Reviewer #2: The current manuscript by Seveno et. al. builds on their previous study on PfPP1’s role in the process of schizont egress. The study investigates the role of PfPP1 using fluorophore-tagged and nano-luciferase reporter lines combined with live microscopy and chemical kinetics. After demonstrating that PP1 deleted parasites are blocked prior to rupture of their PVM, the authors utilize A23187 and Zaprinast/BIPPO to overcome the egress defect. The Ca2+ ionophore appears more effective and quicker to bypass the PP1-deletion block. Overall, the manuscript is nicely done but some of the conclusions are somewhat overstated. The manuscript would be improved by some clarifications discussed below.

Despite the importance of the work and the impressive experiments, some concerns about the manuscript that are listed below.

Reviewer #3: In this paper the role of Protein Phosphatase 1 (PP1) is investigated using a clever combination of molecular genetics, small molecule inhibitors and live cell imagining. Egress is divided into several stages beginning with the schizont having an irregular (scallop shell) appearance with the merozoites partly distributed around the central residual body. The merozoites then fully redistribute around the residual body before PVM breakdown followed by RBC membrane breakdown and merozoite egress. Deletion of PP1 arrested egress at the irregular stage by reducing cGMP synthesis which would normally activate PKG. PKG activation in turn, raises Ca2+ levels and activates CDPKs involved in organelle secretion of proteins involved in breakdown of the PVM and RBC membrane. Egress can be rescued in ∆PP1 parasites via addition of BIPPO which raises cGMP levels and a calcium ionophore. Further experiments indicate that PP1 in addition to the irregular stage also has a role downstream of the PV rounding up stage suggesting it acts at multiple points during egress.

**Part II – Major Issues: Key Experiments Required for Acceptance**

Reviewer #1: (No Response)

Reviewer #2: Major

1. Throughout the manuscript, some percentages and numbers are provide without standard deviation (e.g., line 161, line 163, line 297, line 299). All measurements should have XXX +/- XXX to give the reader some insight about the variability of the measurement.

2. How is the initial time point determined to begin measuring T1, T2, and T4 determined? Is this when the imaging is started or is there is a specific characteristic that defines this time point? In the study, the authors have attempted to calculate the leakage of mRuby3 signal to the host cytoplasm upon PV rupture. However, given the resolution power of the method used, such calculations seem a little precarious.

3. Fig 4: The parasites are treated with C2 which would be expected to block PKG activity and, thus, block PVM poration. Did the authors wash out the C2 before the TEM preparation? The parasites for TEM were treated at 41 hpi and collected at 44 hpi for fixation (lines 569-572), which is likely a few hours before egress. Having said that, the results of electron microscopy showing the absence of PV poration could be considered if the contrast measurement was objective and details were provided. It is currently unclear as to how was the presence or absence of equalization in Figure 4B-C measured.

4. Do the egress inducer-treated parasites reinvade? Can a productive infection be maintained by sequential treatment with BIPPO or A23187 in the PP1 null parasites?

Reviewer #3: I do not have any major issues with this paper.

**Part III – Minor Issues: Editorial and Data Presentation Modifications**

Reviewer #1: For Figure 1

Fig 1A could either include clearer labelling in the figure or describe what the cells represent more thoroughly in the figure legend

General comment - change the word scheme to schematic – this is in multiple figure legends including supplementary figures

Fig 1B – the authors could say how many cells were measured for each condition

Fig 1C – the authors should describe the constructs in this figure legend in the same order as they appear in the figure – the order of the descriptions in the legend does not currently match the order of constructs in the schematic

Fig 1D – the authors could describe what the numbers mean in the top left corner of each image (presumably time – include unit)

Fig 1E – I’m not sure if unpaired t test is the most appropriate statistical test when comparing so many groups. One way ANOVA with post hoc testing might have been a better choice

For Figure 2

Fig 2A – the authors could include fluorescence markers in the labelling on image panels – e.g. PVM-neon green. Indicate a unit for time in the figure legend

Fig 2B-D – more information could be given about exactly how these times are measured – how do they decide when to start measuring? E.g. T1 is a measure of the whole egress process whereas T3 is a measure of just the rupture process. When do you start to measure rupture alone? Is this open to interpretation?

It is also not clear why the authors mention t4 and T5 if they did not give any data on it in this figure. It was given in figure 5 C and D. Can they be either pooled or the T4 and T5 removed from Fig 2 as it leads to confusion.

Figure 3

Fig 3A – the authors could say how many cells were measured

Fig 3A-D – it might be useful to include a label on these graphs showing which parasites these experiments were performed in

In Fig3C, can authors explain that this evidently clear that inhibition by BIPPO has not changed any Ca++ levels in +RAP parasites. It is almost basal. Then how BIPPO dependent Ca++ enrichment is related to any of the results.

Fig 3D – the figure legend says the parasites were treated with A23187 and data normalised to this but this data isn’t on the graph – could it be included for comparison?

Figure 4

Fig 4A – it would be useful to change the colour of the black arrowhead to something which will stand out more

Fig 4C – the data here is only from a single experiment. Could another repeat be added?

Figure 5

Fig 5A – the authors could include a unit for time for the still images in the legend or figure itself.

Fig5C and D can be either pooled to fig 2 or the fig2 and 5 can be put together. The schematic for the different time in Fig2 should be given in Fig5 again so that its clear what t4 or T5 represent.

Line 220: How the measurement of cGMP was performed

Line 224/228: mention fig 3A

Line 232: schematic and fluorescent images of cytosolic ca++ can be detailed in supplementary.

Overall a nice study and few of these changes will improve the manuscript.

Reviewer #2: Minor

1. This is likely something that would be fixed in future iterations, but the locations of the supplemental figures were difficult to determine.

2. The references should be checked for language. I think the dates are in French in some references.

3. Figures labelled as DIC seem to be brightfield images and might be worth checking and correcting.

4. The restoration of PV rounding up in A23187 parasites is also partial (~50%) albeit higher than that of BIPPO (~30%) as per figure 5B. The statement in lines 314 and 315 needs to be edited accordingly.

Reviewer #3: 1. Were any PCRs etc, done to show the loxP flanked PP1 was being deleted efficiently? Given that parasites did not egress Fig 1B, one would assume the deletion was efficient.

2. In Fig 2, the length of time each egress stage lasts are measured. I may have missed this, but how was it determined when to start the measuring? Was this simply done by synchronising the parasites and observing from 43 hpi for example?

3. Line 194 – 195. What is meant by “egress molecules primarily modulate PVM breaching”?

4. Line 200+ . “correlated with an acceleration of T2 (T2: 468 ± 21 sec, ~8 min)”What is being compared to what here because the last two A23187 columns in 2C are NS.

5. A citation to Fig 2D appears to be missing.

6. In Fig 5SC Top, is the red MFI trace referring to mRuby3 or to GCamp6f? This refers to parts D and E too.

7. Line 314. “Importantly, the defect in rounding-up of PfPP1-null parasites (+RAP) was overcome by A23187, while BIPPO only provided a partial complementation (Fig. 5B).” How can A23187 overcome the +Rap effect when it is NS compared to -Rap?

8. In Fig 5A, are the pie charts % or the actual number of cells counted?

PLOS authors have the option to publish the peer review history of their article (what does this mean?). If published, this will include your full peer review and any attached files.

Reviewer #1: **Yes: **Rita Tewari

Reviewer #2: No

Reviewer #3: No

Figure Files:

Data Requirements:

Reproducibility:

References:

---

## [Decision Letter · Decision Letter 1]

17 Dec 2024

Dear Dr. Lamarque,

We are pleased to inform you that your manuscript 'The malaria parasite PP1 phosphatase controls the initiation of the egress pathway of asexual blood-stages by regulating the rounding-up of the vacuole' has been provisionally accepted for publication in PLOS Pathogens.

Best regards,

Christopher J. Tonkin

Guest Editor

PLOS Pathogens

Dominique Soldati-Favre

Section Editor

PLOS Pathogens

Sumita Bhaduri-McIntosh

Editor-in-Chief

PLOS Pathogens

orcid.org/0000-0003-2946-9497

Michael Malim

Editor-in-Chief

PLOS Pathogens

orcid.org/0000-0002-7699-2064

This is a nice study dissecting the role of PfPP1 in P. falciparum egress. The experiments are well controlled and conclusions sound. The authors have done a good job in addressing all the initial reviewer concerns. Upn re-review there is only one very minor point which needs to be addressed.

Reviewer Comments (if any, and for reference):

Reviewer's Responses to Questions

**Part I - Summary**

Reviewer #3: (No Response)

**Part II – Major Issues: Key Experiments Required for Acceptance**

Reviewer #3: (No Response)

**Part III – Minor Issues: Editorial and Data Presentation Modifications**

Reviewer #3: I found the new invasion data in S4 Appendix hard to find as it was inside a supplementary information file. Maybe consider having all the appendix files separate.

PLOS authors have the option to publish the peer review history of their article (what does this mean?). If published, this will include your full peer review and any attached files.

Reviewer #3: No

---

## [Editor Report · Acceptance letter]

22 Dec 2024

Dear Dr. Lamarque,

We are delighted to inform you that your manuscript, "The malaria parasite PP1 phosphatase controls the initiation of the egress pathway of asexual blood-stages by regulating the rounding-up of the vacuole," has been formally accepted for publication in PLOS Pathogens.

Best regards,

Sumita Bhaduri-McIntosh

Editor-in-Chief

PLOS Pathogens

orcid.org/0000-0003-2946-9497

Michael Malim

Editor-in-Chief

PLOS Pathogens

orcid.org/0000-0002-7699-2064